# A Review of Cutting-Edge Sensor Technologies for Improved Flood Monitoring and Damage Assessment

**DOI:** 10.3390/s24217090

**Published:** 2024-11-04

**Authors:** Yixin Tao, Bingwei Tian, Basanta Raj Adhikari, Qi Zuo, Xiaolong Luo, Baofeng Di

**Affiliations:** 1Institute for Disaster Management and Reconstruction, Sichuan University-The Hong Kong Polytechnic University, Chengdu 610207, China; tyx20010518@163.com (Y.T.); zuoqi998@163.com (Q.Z.); luoxiaolong44@163.com (X.L.); dibaofeng@scu.edu.cn (B.D.); 2Department of Civil Engineering, Pulchowk Campus, Institute of Engineering, Tribhuvan University, Lalitpur 44600, Nepal; bradhikari@ioe.edu.np

**Keywords:** floods, sensors, remote sensing, flood losses

## Abstract

Floods are the most destructive, widespread, and frequent natural hazards. The extent of flood events is accelerating in the context of climate change, where flood management and disaster mitigation remain important long-term issues. Different studies have been utilizing data and images from various types of sensors for mapping, assessment, forecasting, early warning, rescue, and other disaster prevention and mitigation activities before, during, and after floods, including flash floods, coastal floods, and urban floods. These monitoring processes evolved from early ground-based observations relying on in situ sensors to high-precision, high-resolution, and high-coverage monitoring by airborne and remote sensing sensors. In this study, we have analyzed the different kinds of sensors from the literature review, case studies, and other methods to explore the development history of flood sensors and the driving role of floods in different countries. It is found that there is a trend towards the integration of flood sensors with artificial intelligence, and their state-of-the-art determines the effectiveness of local flood management to a large extent. This study helps to improve the efficiency of flood monitoring advancement and flood responses as it explores the different types of sensors and their effectiveness.

## 1. Introduction

Flood hazards include flash floods, river overflow floods, storm surge floods, glacial lake outburst floods, and urban waterlogging. In recent years, the frequency and severity of floods have been increasing globally due to the impact of climate change. According to the Global Natural Disaster Assessment Report (2022) [1], flooding has been the most frequent event in the past 30 years, accounting for 38.87% of the total frequency of all disasters where 4321 large or small flood hazards affected a population of 1927.99 million. Geographically, most flood events occur in Asia, Africa, and North America, but more than 50% of these flood losses happened in developing countries around the Himalayas [2]. A recent report published by the Centre for Research on the Epidemiology of Disasters (CRED) shows that flood events always bring huge economic losses, and a large number of people are affected, as was the case when Typhoon Doksuri hit the Chinese coast causing the second most expensive disaster in the world in 2023. South Asian countries are being severely affected by major river systems, such as the Indus, Ganges, Yarlung Tsangpo, and Meghna rivers, which have brought about civilization but caused many severe floods throughout history [3]. Some coastal countries, such as the United States, Japan, the United Kingdom, and Australia, suffer from floods caused by typhoons and heavy rains. Even by late September of this year, the Sahara Desert region was struck by unprecedented torrential rain and flood events, and Lake Iriqui, which had been dry for 50 years, was filled. Climate change is making the global hydrological cycle increasingly unstable and unpredictable. According to the World Resources Institute, floods have caused more than $1 trillion in damages globally between 1980 and 2020 [4]. Previous studies show that the vast majority of the world’s population, those living in low- and middle-income countries, are exposed to flood risk (89%) [5]. It is clear that people in poverty are far more vulnerable to floods than the rich because low-income households often find it difficult to withstand devastating losses from floods [6]. Most flood-prone countries have unique landscapes and climates where different kinds of flood events occur. The rapid, unplanned, and haphazard urban growth increases flash floods or pluvial flooding in many countries [7]. Floods mainly cause casualties and damage to farmland, livestock, and infrastructure, which creates challenges for developing countries. It is very challenging to control the impact of flooding [8], but many efforts have been made globally for flood governance and management, using both structural and non-structural measures [9].

Different flood sensors are used for flood detection and early warning systems around the globe [10]. In recent years, flood sensor technology has made significant progress in terms of hardware, communication, and multi-technology linkages, especially with the emergence of radar remote sensing and social media (Figure 1). Schumann [11] delineated six milestones in flood monitoring from the technical and policy perspectives by sorting out the breakthroughs and advances in satellite remote sensing of floods over the past 50 years. The majority of this research is concentrated on specific regions and particular technologies, with comprehensive studies on flood sensors being notably scarce in the literature. Therefore, this study aims to review the development of global flood sensors and proposes a new classification of flood sensors by selecting representative flood events. This study further analyzes the difference between developing and developed countries in the development and application of flood sensors, focusing on the strengths, limitations, and future directions of different types of sensors in different types of flood scenarios.

## 2. Methodology

### 2.1. Methods and Data Sources

The data were extracted from the Web of Science (WOS) database [12] between 2000 and 2023. This study considered only journal articles, conference papers, and review papers where literature of low relevance to the topic of this study was excluded by analyzing the titles, abstracts, keywords, and other information of the papers for manual screening. The following keywords were used: “flooding disaster/hazard”, “flood monitoring”, “flood risk assessment”, “flood mapping”, “flood forecast”, “flood warning”, “flood recovery”, and “flood disaster/hazard” in the WOS database. The keywords “flood recovery” and “sensor” were used in the Web of Science Core Collection database, yielding 2429 results covering research areas such as remote sensing, telecommunications, oceanography, meteorology, and atmospheric science. After filtering the preliminary search results and excluding irrelevant research topics (e.g., linguistics, stomatology, paleontology), 791 valid literature data on flood sensors were obtained, of which 122 articles were selected for this study. According to the report of the World Resources Institute (WRI) [13], the top fifteen countries in terms of the number of people affected by floods are mostly developing countries or underdeveloped countries, including India, Bangladesh, and China.

Some of the remarkable flood events from 2003 to 2023 show the global distribution of flooding scenarios and the driving role of innovations in sensor technology (Table 1). Overall, the emergence of hardware and software technologies such as wireless sensor networks, satellites, radar, drones, social media platforms, and artificial intelligence is crucial.

### 2.2. Keyword Co-Occurrence Analysis

The co-occurrence analysis shows that the keywords that appear more than 5 times in the literature and the trends of flood sensors in the past two decades were categorized into seven categories (Figure 2). They are: (1) soil moisture: hydraulic modeling, hydrological modeling, flood extent; (2) model: river, uncertainty, optimal selection, flood monitoring; (3) data fusion: deep learning, convolutional neural network, flood forecasting, digital twin, short-term memory; (4) satellite data: SAR, image processing, multispectral index (MSI), multispectral satellite images, urban flooding; (5) wireless sensor networks: network life cycle, disaster management, genetic algorithms, AI; (6) water surface extent: flood impacts, disaster risk management, hazard warnings, flood sensors, benefit-cost analyses; and (7) land surface water index: coverage, enhanced vegetation index (EVI), Landsat, moderate-resolution imaging spectroradiometer (MODIS). Consistent with the analysis of keyword bursts given by CiteSpace (Figure 3), we found that most of the studies used remote sensing techniques to detect flood-prone areas [42], soil moisture [43], vegetation cover [44], and normalized difference vegetation index (NDVI) [45]. The application of wireless sensor networks in flood monitoring and early warning occurred probably around 2004 [46] and rapidly became a research hotspot between 2018–2021. The key terms of flood prediction, flood impact assessment, machine learning, and Landsat in the field of flood sensor applications have appeared in many literatures since 2020 [47,48,49]. The emergence of deep learning and synthetic aperture radar symbolizes a great advancement in machine learning algorithms, as well as physical sensor technologies, which will likely be the keywords of flood sensors in the future.

## 3. Overview of Flood Sensor Development

Flood sensors are used in all stages of flood management (early warning, flood mapping, flood monitoring, flood assessment, and flood recovery). Generally, sensors can be classified into ground-based in situ sensors and space-based sensors [50]. Floods can be categorized by their causes, each with distinct characteristics. We summarized these types and identified suitable sensors for monitoring key indicators to emphasize the importance of sensors in flood management (Figure 4).

In situ and remote sensing sensors are used for flood early warning systems [51]. Common sensors include rainfall, temperature, water level, and flow sensors, which offer more accurate and reliable data than radar and satellite imagery [52]. Remote sensing is essential for precipitation estimation, and ground-based sensors also require a rapid hydrological response to precipitation [53]. Studies have proven that sensor-based river monitoring systems are increasingly important for early warning of river flooding [54] where integration of remotely sensed data can further improve the accuracy and reliability of warnings. Permafrost melting is a major source of water supply for many river and lake systems around the world. The monitoring of snowmelt floods has typically used in situ multispectral sensors to collect data on the water column, but satellite-borne SAR radar altimeters have become a more suitable option due to the limitation of harsh conditions [55]. Coastal floods, such as storm surge floods and tidal floods, are a serious threat to the safety of people and urban infrastructure in coastal areas [56]. Tidal level sensors and wave sensors are widely used in flood warnings, and satellite remote sensing provides an effective method for obtaining large-scale data on sea surface temperature, sea level height, wind speed, and wind direction [57]. Glacial lake outburst floods have a complex mechanism and are usually predicted by four types of indicators: climatic factors, glacier parameters, glacial lake parameters, and dam parameters [58], which rely heavily on satellite remote sensing data for monitoring [59]. Rapidly growing cities are prone to pluvial flooding due to improper urban expansion and inadequate drainage systems. Water sensor-based methods are expensive and susceptible to environmental influences [60,61], whereas remote sensing and social media data can be used for large-scale monitoring and warning instead. Tsunamis are closely linked to seismic activity, and seismographs, along with deep-water tsunami sensors, are key components of the early warning system. The remote sensing sensors do not directly measure tsunami waves, but they can provide communication support and assist in monitoring for early warning [62].

### 3.1. Flood In Situ Sensors

Establishing early warning systems using sensor devices to send alerts via SMS or the Internet is an effective method for reducing flood risk. Different kinds of in situ sensors are used in the early stages of flooding (Table 2). 

These sensors are very important components of the flood early warning for flood level monitoring and detection. There are different methods for flood prediction based on the precipitation data, but they totally depend on the spatial resolution of the dataset [63]. The measurement intervals can be improved in the first place by using automated sensors, such as automated ultrasonic water level sensors [64] and automated pressure sensors. Pressure sensors are more economical, but they require additional barometric measurements to obtain more accurate values. Non-contact radar measurements at times of high water levels are regarded as one of the most accurate and reliable methods, significantly improving temporal and spatial resolution. In addition, fiber optic sensors [65] and capacitive sensors are also used for flood detection. Infrared spectroscopy (IR) sensors are useful for water measurement; however, they have different issues during the measurement process due to the limitations of the principle of operation, the measurement distance, the cost of maintenance, and the environmental disturbances [66]. There are a few urban flood-sensing installations that use IR sensors for the feeder task of remote temperature sensing [67]. Ultrasonic sensors, due to their ability to penetrate various media and lower long-term maintenance costs, are more suitable than IR sensors for harsh outdoor environments, making them a more mature and common choice for water level measurement in flood early warning systems [68].

Most of the effective flood monitoring systems or flood wireless sensor networks use ultrasonic sensors as carriers [69,70,71] using Internet of Things (IoT) technology, wireless sensors, and a new type of battery-less wireless sensor based on radio frequency identification (RFID) [72]. The passive nature of these sensors preserves the RFID range, enabling more accurate hydrological data extraction in flood monitoring. Advances in water level monitoring have led to the development of wireless sensor networks (WSNs) [73], a promising technology that offers near-real-time, automated flood monitoring with less energy consumption, extended data transmission distances, and convenient wireless formats [74]. This is now considered to be one of the main promising solutions for flood monitoring [75]. The interconnection of multiple sensors enables low-cost remote monitoring and real-time data sharing of water conditions [76]. Current flood monitoring relies on data from river-based sensors, including direct water level measurements and indirect indicators like temperature, humidity, rainfall, wind speed, and flow rate, which can calibrate hydrodynamic models. Artificial intelligence algorithms can not only combine data from multiple sensors to make decisions about flood hazard prediction and response [77,78]; they can also be used to determine where to deploy sensors [79] and evaluate sensor functionality [80]. The IoT wireless sensor network in Colima-Villa de Álvarez, Mexico, used for hydrometeorological data collection and flood monitoring, has shown good robustness during extreme events [81]. To address sensor node failures during floods, UAV-integrated WSNs can serve as communication routers, collecting data from rainfall and water level sensors using INS and GPS systems [82].

### 3.2. Flood Airborne Remote Sensing Sensors

Since 2014, the advancements in unmanned aerial vehicles (UAVs) have resulted in the extensive application of airborne sensors for the detection of inundated areas in complex urban landscapes and hazardous, inaccessible regions. This technology has been successfully deployed in flood monitoring and rescue missions, such as during the floods in Uttarakhand, India [83], the state of Oklahoma, USA [84], and Malaysia [85]. It can be seen that sensors mounted on UAVs can help to achieve higher temporal and spatial resolution for flood monitoring, mapping, and detection, and they play a role in the monitoring of different flood indicators (Table 3). Compared to traditional aerial photogrammetry, UAV photography offers improved spatial accuracy and maneuverability during flood surveys [86,87]. In previous years, drones were used to carry optical cameras to capture flooded areas [88] but it has taken a long time since then to address the waterproofing characteristics of drones in heavy rainfall environments and to improve the low quality of the data from raindrop-obstructed images. More recently, computer vision techniques have been applied effectively for monitoring water levels in flood warnings using ground-based fixed cameras [89]. Satellites and space shuttles, while valuable, are often limited by their orbital paths, making it challenging to obtain high-quality images; UAVs, with their ability to capture oblique images, can help fill this data gap [90,91]. These techniques have proven superior in disaster damage assessment and flood mapping, as seen in the 2017 Hurricane Harvey [92] and 2021 Melamchi floods [93]. As technology advances, drones have been recognized as an effective platform for flood management [94]. Some studies used photogrammetric data collection on drones equipped with instruments such as high-resolution cameras, LiDAR, radar, multispectral cameras, and thermal sensors to capture the data [95] to create digital elevation models (DEM) [96] or to employ deep learning algorithms for delineating flood extents [97,98].

### 3.3. Flood Satellite Remote Sensing Sensors

As the performance of sensors continues to improve, satellite remote sensing is gradually providing an alternative over traditional measurements with low costs, more frequent revisit intervals, and wider coverage [99]. It is seen that the emergence of satellite remote sensing has solved the problem of inaccessibility and measurement of flooded areas, and many studies on flood monitoring, assessment, early warning and prediction have been carried out using remote sensing technology. However, it is undeniable that the problem of temporal and spatial resolution has hindered the development of remote sensing technology for a long time [100]. At present, satellite remote sensing is mainly classified into optical remote sensing, infrared remote sensing, and microwave remote sensing. These three types of remote sensing sensors have been increasing in measurement accuracy and pursuing environmental immunity, and their monitoring characteristics apply to different flooding situations (Table 4). The passive multispectral scanner (MSS) on Landsat provided 80 m resolution images for early flood-prone area studies in the U.S. In the 1980s, the Landsat thematic mapper (TM), with a 30 m resolution and an 18-day revisit cycle, became the primary data source for flood monitoring and boundary delineation [101]. Later, the SPOT satellite’s X spectral mode enabled multispectral imaging for small-area flood monitoring. The advanced very high-resolution radiometer (AVHRR) has a low spatial resolution but a high temporal resolution with a large spatial range. These characteristics have significantly improved real-time flood detection and monitoring, particularly in underdeveloped areas, due to its cost-effectiveness and minimal technical requirements. In addition, sensors such as the multispectral imager (MSI), enhanced thematic mapper (ETM+), and operational land imager (OLI) have played an important role in flood management activities, especially MODIS, providing globally consistent and independently verifiable data in transboundary river systems [102]. In the early 21st century, hyperspectral sensors enabled optical satellites to capture a broader range of spectral bands, enhancing land cover classification and environmental monitoring.

Between the 1980s and 1990s, various types of satellite sensors were developed, including optical, radar, and infrared sensors, for different remote sensing missions. Optical remote sensing is highly effective, but its ability to penetrate cloud cover is limited. During flooding events, cloud cover is often dense, particularly during monsoon seasons, which poses significant challenges. Synthetic aperture radar (SAR), however, can penetrate cloud cover, making it more suitable for these conditions [103]. The launch of civil SAR satellites, such as ERS-1, ERS-2, and RADARSAT, contributed to the increased popularity of SAR sensors in flood management applications [104]. The Key Earth observation programs include the German TerraSAR-X and TanDEM-X, the Italian COSMO-Skymed, and the Canadian RADARSAT-2. RADARSAT has proved to have the capability to accurately assess and identify inundated areas and to take images on specified dates and locations, thus enabling successful flood monitoring and mapping [105]. The low resolution of the special sensor microwave imager (SSM/I) limits its use in analysis. In the 21st century, Sentinel-1 became widely used after the launch of Europe’s Copernicus program. The U.S.-Japan Tropical Rainfall Measuring Mission (TRMM), which launched in 2015, marked a milestone in space-based precipitation estimation, later succeeded by the higher-resolution Global Precipitation Measurement Mission (GPM).

The use of LiDAR data with sub-meter spatial resolution has proven valuable for urban modeling; however, it is more expensive compared to SAR imagery. LiDAR effectively identifies vertical differences in terrain and flood depths, making it suitable for flood mapping in flat terrains. Furthermore, LiDAR can compensate for the side-viewing nature of SAR for urban buildings and vegetation. For economic reasons, the DEMs of interferometric synthetic aperture radar (IfSAR) and SAR are being used instead of the LiDAR system for large-scale monitoring in many areas [106]. Many organizations independently use various types of remote sensing data for flood monitoring, mapping, prediction, and assessment. However, combining data from different sensors can offset their individual limitations [107,108,109]. The integration of radar and multispectral imagery has demonstrated a marked enhancement in the delineation of flood extents, particularly in conditions characterized by cloud cover [110,111]. Within recent years, machine learning algorithms have facilitated the robust fusion of disparate data sources, leading to significant advancements in the estimation of flood water bodies, assessment of flood risks, and the spatial mapping of flood events [112,113]. It is worth noting that as technology in the field of remote sensing matures, there is a growing consensus to enhance satellite mission support for flood relief efforts [114].

The remote sensing satellite missions based on the sensor, principle, resolution, and launching country (Table 5) shows that there is a gradual transition of satellite missions from initial military reconnaissance to civil and scientific uses and from initial optical sensors to more advanced SAR and hyperspectral sensors. Nearly 50% of those satellites had been launched independently or with the participation of the United States. A number of countries in Europe had the capability to launch satellites, while in Asia, China and India had launched some of the relevant satellites. It was only in the past two years that China had launched its first commercial SAR satellites.

Some of the satellite data in the table above are completely free of charge: Landsat, TRMM, GCOM-W1, ERS, CloudSat, METOP, CALIPSO, Envisat-ASAR, GPM, SMAP, SWOT, NISAR, as well as MODIS data on the Terra and Aqua satellites, and the data of ESA Sentinel-1, Sentinel-2 are also publicly available through the Copernicus program. Ikonos, World View, Quick Bird, Geo Eye, Qilu 1, Haisi 1, Chaohu 1, NovaSAR-1, TerraSAR-X, Jilin-1, Zhuhai-1, SPOT-6, SPOT-7, RADARSAT-2 are commercial satellites that require payment, but other satellites are partially free of charge (Figure 5). Some of the data sources from early-launched military satellites are old and may not be available for future use; however, once declassified, these data are valuable for generating and reconstructing historical records of disasters and disaster events.

### 3.4. Flood Social Sensors

Social sensor data has already demonstrated its value in densely populated city-level flood monitoring and early warning. Citizen crowd-sourcing has been proposed as an alternative to adaptive data collection in the context of the widespread use of advanced sensor technologies and wireless sensor network for flood management [115], where individuals can provide volunteered geographic information (VGI) [116]. These data are particularly important in areas where there is no monitoring. Wang [117] demonstrated that data collected from Twitter and MyCoast (a crowdsourcing application) can complement super-resolution datasets for urban flooding. Huang [118] combined real-time flow sensor data, social media tweets, and remotely sensed imagery to create a near-real-time flood inundation model using the normalized difference water index (NDWI). Data from travel navigation platforms, including crowdsourced traffic and flood event data, can be used to estimate flood impacts on traffic [119]. Citizen science-based integration of flood information to estimate flood extents in areas with limited terrestrial sensors can complement physical sensor measurements. Maurizio [120] demonstrated the usefulness of assimilating crowdsourced observations from a heterogeneous network of static physical, static social, and dynamic social sensors. During Hurricane Harvey in 2017, crowdsourced data and other information were used to predict the probability of flooding, validate model performance, and improve flood extent estimates based on 311 flood call information [121]. Although big data from different physical sensors are widely available but they cannot be used alone in many cases to meet (near) real-time data because of spatiotemporal resolution. The distributed data fusion framework proposed by Li J [122] integrates remote sensing and social media data, which shows high processing efficiency and potential for fast computation in real-world scenarios. In addition to solving the problem of data quality through data fusion, machine learning algorithms also offer new solutions for efficient information extraction from social media data [123,124], but due to API limitations, current research mainly relies on Twitter datasets. Unfortunately, Twitter has removed the ability to freely access tweets.

In practice, data on physical disaster attributes and social crowdsourcing are complementary, particularly in underdeveloped and disadvantaged areas, where they compensate for sparse physical sensors and unreliable data. However, these regions often have outdated mobile devices and sparse populations, raising concerns about the accuracy of social media data. Therefore, integrating physico-social sensor data is essential for effective flood disaster response.

### 3.5. Flood Datasets

With the advancement of big data and artificial intelligence, flood disaster management has expanded into new research areas. Supported by various sensor technologies, numerous publicly available datasets have been created to provide foundational information for flood management studies. These datasets have been systematically compiled by scholars such as Zhao Jie [125,126], and building upon this work, we have categorized flood-related datasets into three primary types: single-type datasets, multimodal datasets, and text-image datasets (Table 6). The majority of these datasets focus on the flood event itself, while a smaller portion captures related indicators such as roads and buildings. Satellite imagery predominantly originates from Sentinel-1 and Sentinel-2. Due to the uneven global distribution of social media users and the diversity of languages and posting formats, unified social media datasets related to floods remain limited. However, in recent years, some researchers have developed datasets based on platforms such as Twitter [127]. Additionally, datasets gathered from physical sensors, such as Sen1Floods11, Sen12-Flood, and FloodNet, are widely recognized by the research community. In contrast, xBD is less commonly used due to its specialized data focus and registration requirements. Furthermore, free satellite datasets containing flood hazard information are available from platforms such as USGS Earth Explorer, Sentinel Copernicus Browser, and NOAA Data Access Viewer. These include data from Landsat 1–9, Sentinel-2, TerraSAR-X, MODIS, AVHRR, and other sensors, all of which are instrumental in flood research and monitoring.

## 4. Global Applications of Flood Sensors

Developed countries exhibit some differences from developing countries in terms of flood sensor technology and its applications. To preserve spatial variability and disregard the temporal stratification of event occurrence, we searched and filtered significant flood events with substantial mortality and economic loss during the same period in the EM-DAT database for comparison. We explored the link between the application of flood sensors and flood disaster losses to substantiate the importance of flood sensor development in flood management (Table 7). What is currently known is that developed countries have state-of-the-art hydrometeorological analysis and proximity forecasting products and use more advanced processing methods for flood data [128,129]. Developing countries, on the other hand, generally have poor flood defense infrastructure [130], and citizen science has been proven, in practice, to be one of the most useful methodology applied to manage flood risk in the poorest, most vulnerable, and marginalized areas [131], but it has limited data volume, accuracy, and emergency response capacity, with the IAHS reporting that developing countries often suffer greater flood losses due to the inability to make accurate flood forecasts [132]. In addition, there was still a serious knowledge gap between developing and developed countries in terms of data utilization, potential, and challenges [133], with developed countries having a wider range of satellite research activities compared to developing countries [105], with timely access to a large amount of data through domestic space agencies, whereas, on the contrary, most developing countries have difficulty in accessing it from domestic sources [134] and there is a large time lag in obtaining satellite resources through open access, which affects the real-time monitoring and relief of floods.

Natural hazards such as flash floods, landslides, and mudslides are always frequent in developing countries around the Hindu Kush Himalayan (HKH) region [135], among which Pakistan suffers from floods almost every year. The floods of 2024, 2022, and 2010 severely affected the Khyber Pakhtunkhwa, Punjab, and Balochistan areas, with the worst floods in Pakistan’s history occurring in 2010, when the country suffered a death toll of 1985 and economic losses of up to $10 billion, with poor flood protection infrastructure in numerous areas of the territory and little to no early warning system, even in the north-western and coastal areas of Balochistan [136]. Lower Punjab and Sindh also suffered from the lack of complete emergency response and post-disaster relief systems, which made it difficult to access outside assistance [137]. In addition to poor infrastructure, less advanced warnings, poor data processing technology [138], and slow emergency response [139], which are typical reasons for the failure of flood response in the country, institutional corruption, mismanagement of water resources, and lack of an effective flood policy exacerbated the severity of this flood problem [140,141]. The 2011–2012 floods in Thailand, also caused by monsoon rains, affected the lives of more than 13 million people in 66 provinces and resulted in 813 deaths and US$50 billion in economic losses, making it the worst flood disaster in Thailand’s history, with disruptions in the supply chain of the global automotive, electronics, computer hardware, and other industries, which triggered academic attention to the resilience of the supply chain to floods [142]. During the floods, in situ sensors, social sensors, and satellite remote sensing sensors are not utilized in the most effective way, especially in urban areas in bad weather [143], and unscientific land-use planning and poor-quality flood defenses directly exacerbate the impacts of the floods [144]. A study of Thailand’s flood resilience building, published in 2018, noted that Thailand more than likely continued to opt for constructive measures to mitigate the floods after the 2011 floods [145], but in reality, the Chao Phraya River floods caused such high casualties and economic losses, imperfect national institutions and flood policies also need to bear much of the blame.

On the contrary, the performance of developed countries in the face of large-scale floods has demonstrated significant advantages in flood prevention infrastructure, early warning systems, social media messaging, and the use of ground-based and telemetry data, etc. The 2010–2011 floods in Queensland, Australia, were one of the worst natural disasters in its history, affecting the daily lives of more than 170,000 people. Thanks to the timely and accurate ALERT flood warning system, evacuation measures, and Facebook disaster information dissemination [146], the actual number of deaths (35) was kept at a low level, while 25 flood-scoping products were developed by collecting data from various parts of the affected area, as well as more than 600 satellite images and derived data obtained from the US, UK, and Germany, which compensated for the lack of data from ground-based sensors and helped the smooth management of floods [147]. Learning from the lessons of its “Flood of the Century” in 2002, Germany invested heavily in the development of constructive and non-constructive measures that played a key role in the 2013 flood event. The International Charter for Space and Major Disasters was activated in the immediate aftermath of the floods, and with the help of meteorological radar and Earth observation satellites, the Deutscher Wetterdienst (DWD) issued timely and accurate warnings that were continuously updated to the public and relevant organizations in virtually all regions affected by the floods. Social media and drone technology [148] also played a significant role in the flood management process. Statistics indicate that, ultimately, only 5% and 3% of companies were affected by not receiving any flood warnings [149].

## 5. Conclusions

The monitoring and damage assessment of floods by using different kinds of remote sensing and ground-based sensors highlighted the influence of climate change, flood events, and technological advancements on the evolution of flood sensor technologies and policies. Since 2000, global warming has driven a rise in extreme weather events, increasing the demand for effective flood management systems. It is found that a strong correlation exists between the advancement of flood sensor technology and the effectiveness of flood management. In situ sensors, evolving with IoT support, remain fundamental, while remote sensing and social sensors show promise but require further development in cost-efficiency and data accuracy. Due to the high cost of precision sensors and platforms, even developed countries are constrained from deploying them extensively.

AI and multi-source data integration offer significant potential, particularly in the area of flood monitoring, forecasting, and management. AI can help collect, analyze, and publish data from a variety of sources provided by sensors, such as remote sensing, smart sensors, and social media, and machine learning and deep learning models, such as support vector machines (SVMS), random forests (RF), convolutional neural networks (CNN), long short-term memory networks (LSTM), U-Net, and Transformer can play a different role when combined with historical or real-time data and can be used to improve the flood warning and forecasting system and various evaluation systems to improve the efficiency of flood disaster management. Satellites have become an effective tool to acquire global data sets, so extensive research on artificial intelligence algorithms to process integrated spatial data has become an inevitable trend, especially in the more complex, image-based flood monitoring, in which the remote sensing platform reflects the economy and efficiency.

Recent flood monitoring studies indicate that developing countries face significant challenges in managing floods. In addition to environmental factors such as complex terrain, topography, vegetation cover, and lighting conditions, their inadequate sensor deployment, emergency response mechanisms, and experience in flood management further hinder effective disaster management. The most direct solution would be to invest in flood detection infrastructure and technology, develop multi-source data integration, enhance data analysis capacity, and update flood policies and laws. Also, urban flooding is different from other types of flooding, requiring more advanced sensor technology and data processing technology. However, if developing countries struggle in the short term to deploy high-precision and high-density sensors or train skilled personnel in advanced remote sensing technologies, they can seek assistance through international cooperation programs or organizations such as Sentinel Asia, IFI, GFDRR, etc. For example, during the 2010 Pakistan floods, the country, as a member of the program, accessed satellite imagery from ALOS, Resourcesat-1, and FORMOSAT-2 to support disaster management. Based on this, we propose a framework to facilitate global collaboration and resource-sharing in flood disaster management (Figure 6).

In light of the current situation, developing countries should prioritize structural measures, emphasize the training and recruitment of technical personnel, and avoid the scenario of possessing basic data without the capacity to utilize it effectively. Additionally, enhancing disaster awareness and skills is crucial to strengthen the self-rescue capabilities of residents. Developed countries should consider prioritizing the development of non-structural measures for a safer and more resilient future for everyone exposed to the threat of flooding.

## Figures and Tables

**Figure 1 sensors-24-07090-f001:**
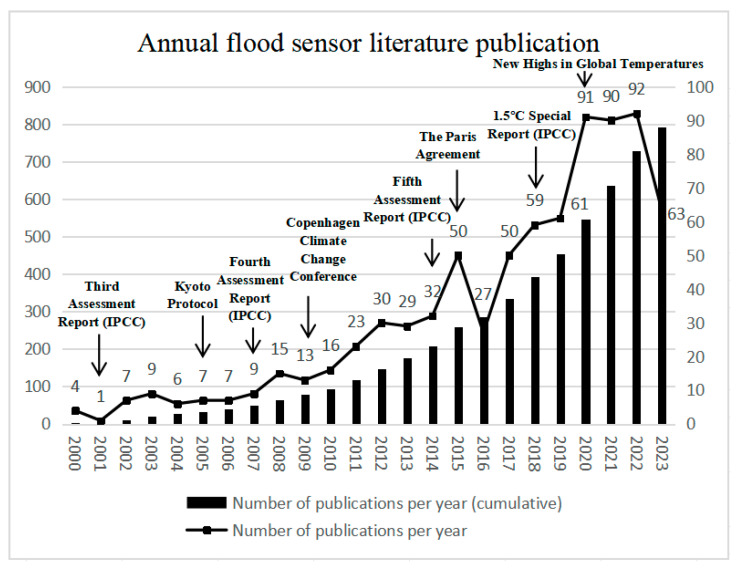
Annual flood sensor research literature publication.

**Figure 2 sensors-24-07090-f002:**
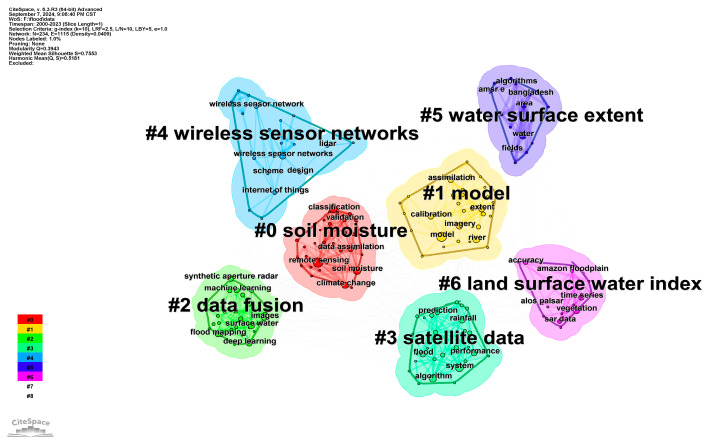
Co-occurrence of keywords about flood sensors in WOS.

**Figure 3 sensors-24-07090-f003:**
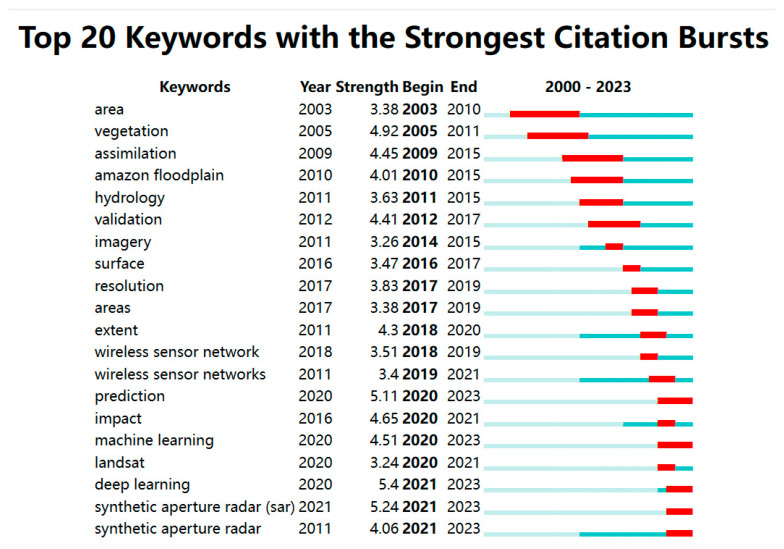
Burst analysis of flood sensor keywords. (The graph is sorted by the start of the burst, with dark blue indicating the timeframe in which the keyword appeared, and red indicating the keyword with the highest citation intensity in a given time period).

**Figure 4 sensors-24-07090-f004:**
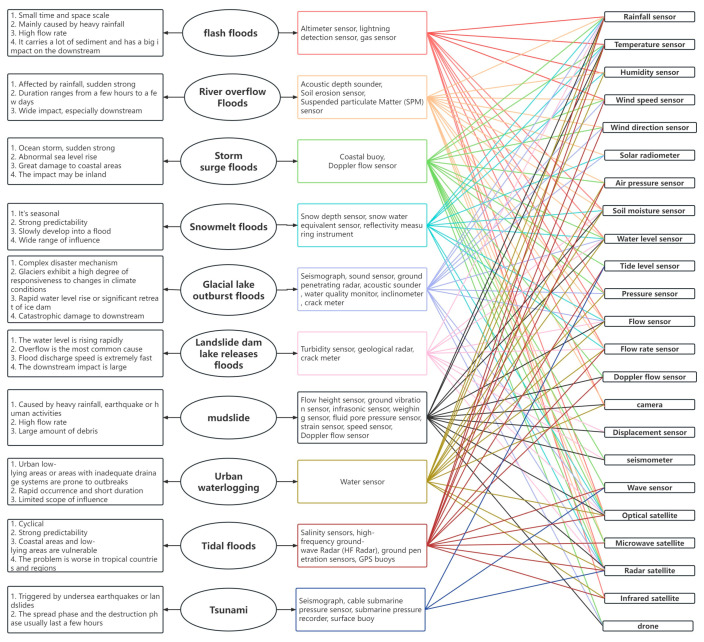
Summary of early monitoring sensors for various types of floods.

**Figure 5 sensors-24-07090-f005:**
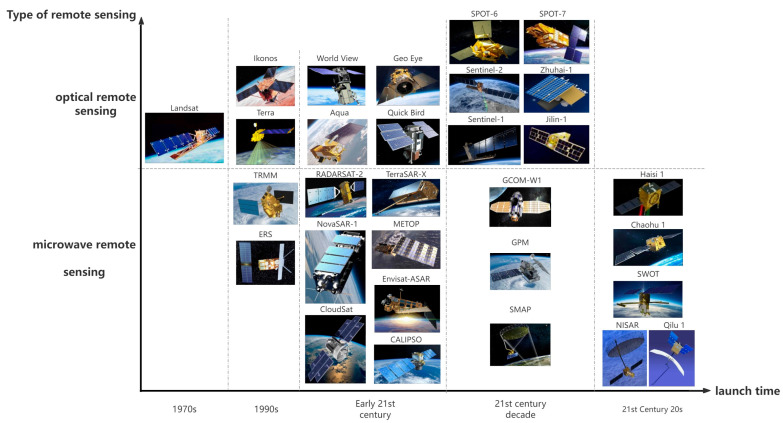
Totally free and non-free flood management satellites.

**Figure 6 sensors-24-07090-f006:**
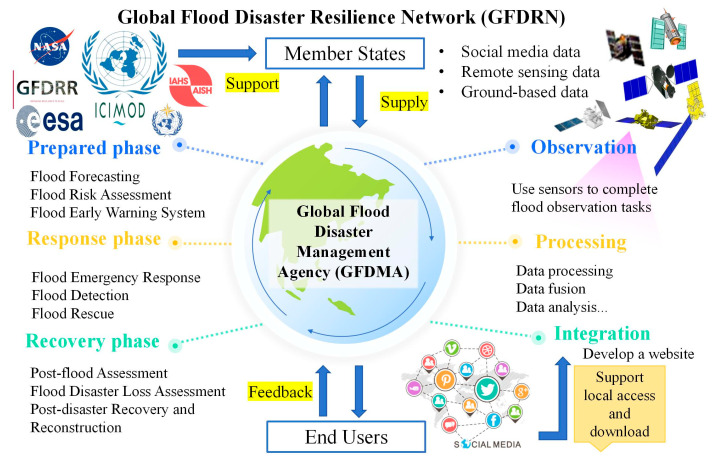
Global collaborative framework for flood disaster management.

**Table 1 sensors-24-07090-t001:** Flood events drive flood sensor development.

Year of Occurrence	Flood Event	Flood Sensor Research Directions
2000	Mozambique Floods	Remote sensors respond to floods in remote areas [14]
2002	Elbe and Danube Floods	Emerging technologies for flood monitoring and early flood warning systems [15,16]
2005	Hurricane Katrina Floods	Satellite radar and airborne remote sensors for tracking and monitoring floods [17,18,19] Advanced sensor data processing techniques [20]
2007	Summer Floods in the United Kingdom	Weather radar and wireless sensor networks help to extend flood warning time [21,22]
2010	Pakistan floods	Integration of multi-source sensor data, including in situ and remote sensing, to improve flood monitoring systems [23,24] Low-cost wireless sensor networks and communication technologies [25]
2011	Thailand floods	Sensor networks integrating satellite data to provide real-time or near-real-time flood monitoring [26,27]
2011	Tohoku earthquake and tsunami, Japan	Social media sensors for flood warning and response [28,29]
2013	North India floods	High-resolution satellite data for flood warning and emergency management [30,31]
2013	Elbe and Danube Floods	Improved sensor accuracy and speed for flood response [32,33,34]
2017	Houston Floods	Multi-radar multi-sensor system for urban flooding [35]
2018	Kerala floods, India	High-time and high-resolution meteorological remote sensing data [36,37]
2019	Mumbai floods, India	Smart flood sensors [38,39]
2022	Pakistan floods	Integrating artificial intelligence and machine learning for flood data processing [40,41]

**Table 2 sensors-24-07090-t002:** Advantage and disadvantage analysis of flood in situ sensors.

Sensor	Index	Advantages	Disadvantages
Pressure Sensors	water level, water pressure	1. High-precision 2. Real-time 3. Deep water and submarine environment 4. Anti-interference	1. Requires regular calibration 2. Stable power supply is required 3. Restricted installation locations 4. Subject to environmental influences
Electromagnetic Sensors	water level, flow rate, velocity	1. Non-contact measurement 2. High accuracy 3. Suitable for different media	1. High costs 2. Need for a stable power supply
Capacitive Sensors	rainfall, humidity, barometric pressure, soil moisture, water level, water quality, displacement, depth of standing water	1. High-precision 2. Suitable for different media 3. Anti-interference	1. Periodic calibration required 2. Subject to environmental influences
Resistive Sensors	humidity, soil moisture, water level, water quality, displacement, barometric pressure, depth of standing water	1. Simple structure 2. Economical	1. Lower accuracy 2. Easy to be corroded
Fiber Optic Sensors	temperature, water level, water quality	1. Compact structure 2. High flexibility 3. Remote monitoring 4. Anti-electromagnetic interference	1. High costs 2. Complex installation and maintenance techniques
Ultrasonic Sensors	wind speed, wind direction, water level, flow rate, flow velocity, wave height	1. Non-contact measurement 2. High-precision 3. Fast response time 4. Unaffected by the environment	1. High costs 2. Easily damaged 3. Affected by temperature/noise 4. Limited sensing distance
Infrared Sensors	temperature, water level, water quality, wave height	1. Non-contact measurement 2. Fast response time 3. Economical 4. Particularly suitable for flat and smooth water surfaces	1. Affected by temperature/noise 2. Affected by the surface properties of objects 3. Higher accuracy within a certain spatial range
Acoustic Sensors	rainfall, rain sound, underground sound, water levels, underwater velocity, underwater topography, underwater objects	1. Detectable underwater conditions 2. Adaptable to harsh environments	1. Affected by the quality of water 2. Complicated data processing
Wave Sensors	water level, flow velocity, flow direction, underwater objects, ground vibration	1. Real-time 2. Remote within a certain range	1. Affected by temperature/noise 2. High costs 3. Complicated data processing
Float Sensors	water levels, flow velocity, fluctuations	1. Simple and reliable 2. Economical 3. Easy to install and maintain	1. Low-precision 2. Easily damaged 4. Not suitable for high-pressure environment
Buoy Sensors	water levels, water quality, flow velocity, waves, flood extent	1. Highly mobile 2. Adaptable	1. High costs 2. Low positioning accuracy
Line Sensors	water levels, flow velocity	1. Economical 2. Suitable for confined spaces	1. Easily damaged 2. Restricted mounting position
Camera Sensors	water levels, water quality, wave heights, fluctuation, flood extent	1. High resolution 2. Intuitive visual information 3. Real-time	1. Visibility is easily affected 2. Restricted installation location 3. Complicated data processing
Bio Sensors	water quality	1. Sensitivity to environmental changes	1. Vulnerability to environmental interference 2. Complex data processing
Microwave Sensors	water level, flow velocity, submerged objects, depth of standing water	1. High-precision 2. Real-time 3. Remote monitoring 4. Not affected by the weather	1. High costs 2. Vulnerable to interference
Radar Sensors	water levels, flow velocity, wave height, underwater objects	1. Remote monitoring 2. Highly adaptable	1. High costs 2. High power consumption
Ground LiDAR	water Levels, wave height, fluctuations, flood extent, flood depth, topography, deformation of geological structures	1. Small-area economy 2. Portability 3. High-resolution, high measurement density, high-precision 4. Unaffected by light	1. High costs 2. Long data processing time

**Table 3 sensors-24-07090-t003:** Advantage and disadvantage analysis of flood airborne remote sensing sensors.

Sensor	Index	Advantages	Disadvantages
Optical Camera	surface features, color	1. High-resolution 2. Intuitive and easy 3. Low costs	1. Affected by weather 2. Limited performance at night or in low-light conditions
Camera	topographic changes	1. Real-time 2. Intuitive dynamics	1. Affected by light and weather 2. Complexity of data processing
Infrared Cameras	water body temperature, ground temperature, vegetation health	1. Capable of monitoring temperature and water flow 2. Can be used at night or in low light conditions	1. Low-resolution 2. Affected by weather conditions 3. Not suitable for wide-range target identification
Multi-spectral Cameras	vegetation type, vegetation health, water body boundaries, flood extent	1. Analyzing specific bands 2. Capable of differentiating between vegetation and water bodies	1. High costs 2. Complex data processing 3. Limited resolution and spectral range 4. Unsuitable for complex features
Hyperspectral Cameras	water quality, vegetation health	1. High spectral resolution 2. High-precision 3. Detailed surface analysis	1. High costs 2. Large volume of data 3. Complex data processing
Thermal Infrared Sensors	water body temperature, ground temperatures, flood extent, changes in humidity	1. Suitable for wide-area monitoring 2. Can be used at night	1. Low-resolution 2. Low-precision 3. Vulnerability to environmental interference
Passive Infrared Sensors	moving objects, thermal radiation	1. Usable at night or in low visibility 2. Monitor animals/vehicles	1. Monitor within a small range 2. No exact distance information 3. No detailed image information
Ultrasonic Sensors	water level, object distance	1. High-precision 2. Low costs 3. Direct measurement	1. Affected by weather and environment 2. Limited installation location 3. Monitor within a small range
Passive Microwave Radiometers	atmospheric humidity, soil moisture, flood extent	1. Suitable for all-weather monitoring 2. Can penetrate clouds and some vegetation	1. Low-resolution 2. Unsuitable for measuring small water body boundaries 3. Complicated data processing
Radar Altimeter	ground/sea level	1. High-precision 2. Unaffected by light	1. Affected by the atmosphere 2. Limited measurement range
Optical Radar	3D terrain, object shape	1. High spatial resolution 2. Precise 3D information	1. Higher costs 2. Subject to weather conditions
Laser Radar (LiDAR)	water levels, vegetation structure, flood depths, surface elevations, terrain changes geomorphological changes	1. High-resolution 2. Unaffected by light and cloud cover 3. Unaffected by terrain shadows 4. Fast data acquisition 5. Highly automated	1. High costs 2. Can be used in extreme weather 3. Not well representative of small-scale terrain features 4. Affected by vegetation cover
Interferometric Radar (InSAR)	terrain changes	1. High-precision 2. Suitable for detecting small changes	1. Complexity of data processing 2. Multiple data calibrations required
Interferometric Synthetic Aperture Radar (IfSAR)	3D terrain, terrain change	1. Flexible system deployment 2. High spatial resolution	1. High costs 2. Complex data processing 3. Limited in urban areas 4. Affected by vegetation cover
Synthetic Aperture Radar (SAR)	water table, surface moisture, surface features, terrain	1. High-resolution 2. Unaffected by light and weather conditions	1. High costs 2. Complex data processing 3. High reflectivity to the water surface
Positioning Sensors	location	1. Precise geographic location information 2. Navigation aids	1. Requires good communication support (e.g., GPS) 2. Not suitable for remote or disaster-prone areas
Orientation Sensors	location, trajectory, attitude, direction	1. Usable at night or in low visibility 2. Data calibration assistance	1. Requires calibration 2. Dependent on positioning 3. Not suitable for stand-alone use 4. Interfered with environmental factors such as magnetic fields

**Table 4 sensors-24-07090-t004:** Advantage and disadvantage analysis of flood satellite remote sensing sensors.

Sensor	Measuring Range	Advantages	Disadvantages
Multispectral Sensors	Meter level	1. High spatial resolution 2. Multispectral 3. Recognition of surface cover types	1. Affected by clouds and atmosphere 2. Not real-time 3. Unsuitable for measuring small-scale floods
Hyperspectral Sensors	Meter level	1. High-resolution 2. Accurately identify surface changes	1. High costs 2. High volume of data 3. Complex data processing
Ultra-spectral Sensors	Meter level	1. Extremely high-resolution 2. Detectable weak changes	1. High costs 2. High technical requirements
Near Infrared Sensors	Meter level	1. Can distinguish between water bodies and vegetation	1. Affected by clouds and atmosphere 2. Requires atmospheric correction
Mid-infrared Sensors	Meter level	1. Can be used at night 2. Sensitive to water and ground temperatures	1. Affected by clouds and atmospheric humidity 2. Low-resolution
Far Infrared Sensors	Meter level	1. Suitable for monitoring surface temperature and atmospheric changes	1. Accuracy is affected by the environment 2. Requires atmospheric correction
Passive Microwave Radiometer	Meter level	1. Unaffected by light and weather conditions 2. Ability to penetrate clouds 3. High spatial and temporal resolution	1. High costs 2. Accuracy affected by environment and ground conditions 4. Unsuitable for high water content
On-board Advanced Synthetic Aperture Radar (ASAR)	Meter level	1. Suitable for all weather conditions 2. Insensitive to terrain and ground cover	1. Low-resolution 2. Limited coverage
Radar Altimeter	Centimeter level	1. High-precision 2. Suitable for measuring changes in water surface height	1. Affected by terrain 2. Limited application range 3. Unsuitable for dense vegetation areas
Interferometric Synthetic Aperture Radar (IfSAR) on Board	Sub-meter level	1. High-precision DEM 2. Can recognise terrain changes	1. High costs 2. Complicated data processing 3. Limitations in urban areas
Satellite-borne Synthetic Aperture Radar (SAR)	Sub-meter level	1. Can penetrate clouds and part of the vegetation 3. Suitable for wide-area monitoring	1. High costs 2. Complicated data processing
Satellite LiDAR	Sub-meter level	1. High-precision 2. Accurate monitoring of terrain changes	1. High costs 2.Blocked by clouds 3. Unsuitable for large-scale mapping

**Table 5 sensors-24-07090-t005:** Summary of existing satellite remote sensing sensors for floods.

Satellite/Mission	Sensor	Principle	Resolution (m)	Launching Country
Corona	KH-1, KH-2, KH-3, KH-4A/4B	Optical	0.6–3	America The early 1960s
Argon	KH-5	Optical	Sub-meter level	The Soviet Union The early 1960s
Lanyard	KH-6	Optical	Sub-meter level	America The early 1960s
Gambit-1	KH-7	Optical	Sub-meter level	America The early 1960s
Gambit-2	KH-8	Optical	Sub-meter level	America Middle 1960s
Hexagon	KH-9	Optical	Sub-meter level	America The late 1960s
Landsat	MSS, TM ETM+ OLI, TIRES	Optical	15–30	America The early 1970s (debut)
IRS	PAN LISS WiFS	Optical	5.8 23.5 188	India The late 1980s (debut)
Terra	ASTER	Optical	15–30	America The late 1990s
Terra, Aqua	MODIS	Optical	250–1000	America The early 2000s
Sentinel-2	2A,2B	Optical	10–20	ESA 1910s
Sentinel-1	1A,1B	Microwave	5–25	ESA 1910s
ATS	IRSS XRS, UVS	Optical	Hundreds–Thousands	America 1960s (debut)
DMSP	SSM/I SSM/IS SSM/T SSM/T-2 OLS, SEM	Optical	Hundreds–Thousands	America 1960s (debut)
Bhaskara-1	SAMIR, MRR, HAIMRT, CR	Microwave	80	India The late 1970s
Bhaskara-2	SAMIR MSS	Optical Microwave	40	India The early 1980s
GEOS	ABI, GLM AMSR SEISS	Optical	500–2000	America 1970s (debut)
GMS	VISSR	Optical	1250–5000	Japan The late 1970s (debut)
SPOT	HRV, HRG HRP HOVER	Optical	1.5–20	French 1980s (debut)
NOAA-11	AVHRR HIRS, MSUSSM/I TOVS	Optical	1000–4000	America The late 1980s
Fengyun-1	VISR	Optical	1100	China The late 1980s (debut)
JERS-1	VNR, SWIR SAR	Optical Microwave	18–24 10–100	Japan 1990s
Fengyun-2	VISR, AHI SEM	Optical	1000	China The late 1990s (debut)
Himawari	AHI, SEDA	Optical	500–1000	Japan The late 1990s (debut)
Ikonos	OSS	Optical	1	America The late 1900s (debut)
World View	MSI, PAN SSI, CAVIS	Optical	0.3–0.5	America The late 1900s (debut)
Quick Bird	MSI	Optical	0.61	America The early 2000s
Geo Eye	MSI, PAN	Optical	0.41	America The early 2000s (debut)
ALOS	AVNIR-2 PRISM PULSAR	Optical	2.5	Japan The early 2000s (debut)
Fengyun-3	MWTS IRAS SEM, ERBE LIDAR	Optical Microwave Laser Radar	100– Tens of thousands	China The early 2000s (debut)
Jilin-1	HSI	Optical	0.72–3.24	China 2010s (debut)
Zhuhai-1	HSI	Optical	0.9–3.2	China 2010s (debut)
Gaofen	PMS, HSI AGRI C-SAR X-SAR	Optical Microwave	0.8–500	China 2010s (debut)
TRMM	PR, TMI VIRS, LIS CERES	Microwave	Thousands–Tens of thousands	America-Japan The late 1990s
CloudSat	CPR	Microwave	Thousands–Tens of thousands	America The early 2000s
JASON	CPR	Microwave	0.13	America-France The early 2000s (debut)
RISAT	SAR	Microwave	1–50	India The early 2000s (debut)
METOP	MOS/SMR	Microwave	Hundreds–1000	ESA and EUMETSAT The early 2000s
SRTM	C-SAR	Microwave	10–30	America The early 2000s
GCOM-W1	AMS(R)2	Microwave	Thousands–Tens of thousands	Japan 2010s
GPM	DPR, GMI	Microwave	250–500	America-Japan 2010s
NOAA-20	VIIRS, OMPS CrIS, ATMS	Optical Microwave	375–750	America 2010s
SSC	SIR-A SIR-B SIR-C	Microwave	15–45	America 1980s
RADARSAT	SAR	Microwave	1–5 3–30 30–100	Canada The 1990s (debut)
ERS	SAR, WS ATSRMWR	Microwave	30	ESA The 1990s (debut)
Envisat- ASAR	ASAR	Microwave	30–1000	ESA The early 2000s
TerraSAR-X	X-SAR	Microwave	1–2 2–40	DLR and ESA The early 2000s
COSMO- SkyMed	SAR	Microwave	1–100	Italy The early 2000s (debut)
CALYPSO	IIR CALYPSO	Laser Radar	30–60	America-France The early 2000s
SMAP	L-SAR	Microwave	Meters–Tens of meters	America 2010s
Gaofen	WFV, P/MS C-SAR HRI, MSI AEMS, PSC LiDAR	Optical	0.8–8	China 2010s (debut)
TanDEM-X	X-SAR L-BR S-BR	Microwave	1	Germany 2010s
NovaSAR-1	X-SAR	Microwave	6–30	British 2010s
FSSCAT	GNSS-R/RO/ SCAT HTI	Optical Microwave	Meters–Tens of thousands	ESA The late 2010s
SWOT	RA Ka-RI AMR PODS	Microwave	0.01–0.02 10,000–15,000	NASA and CNES The early 2020s
NISAR	L-SAR S-SAR	Microwave	6	America-India The early 2020s
Qilu 1	Ku-SAR	Microwave	0.5	China The early 2020s
Haisi 1	C-SAR	Microwave	1	China The early 2020s
Chaohu 1	SAR, WTS PHS, TS DO Sensor	Microwave	1	China The early 2020s

**Table 6 sensors-24-07090-t006:** Publicly available flood-related datasets.

Data Type	Datasets	Content
Single-type	Sen1Floods11	Satellite imagery of 11 flood events across the world between 2017 and 2019
	xBD	Satellite imagery with labeled building damage (United States, India, Nepal, and Bangladesh)
	NASA Sentinel-1	Satellite imagery of 5 flood events between 2017 and 2019
	S1S2-Water	65 triplets of Sentinel-1 and Sentinel-2 images with quality checked binary water mask
	S1S2-Flood	Satellite imagery of areas that suffered a major flooding event during the 2019 winter
	Sen12-Flood	Satellite imagery of flood events between 2018 and 2019 (East Africa, South-west Africa, the Middle-East, and Australia)
	UrbanSARFloods	Satellite imagery of 18 flood events across the world
	Roadway flooding image	Visual images of road conditions during flood events used to train road flood prediction models
	Flood Area Segmentation	290 remote sensing images and self-annotated masks for flood area segmentation
	FloodNet	High-resolution drone images of the affected areas after Hurricane Harvey
Multimodal	RAPID-NRT	Satellite imagery and ancillary data (topography data, water occurrence data, hydrography information, etc.) on flooding in the contiguous United States
	MM-Flood	Satellite imagery of 95 flood events in 42 countries, as well as DEM and hydrological maps
	Hurricane Harvey Floods	Rainfall data and flood impact data from Hurricane Harvey
	California flood dataset	Satellite imagery, InSAR data, and optical data of flooding events in California
	MSAW	Geospatial data including SAR and electro-optical images of the Rotterdam area in the Netherlands
	WorldFloods dataset	Multimodal data, such as satellite imagery of 119 flood events between 2015 and 2019
	S1GFloods Dataset	Satellite imagery of 46 flood events between 2015 and 2022
	EU Flood Dataset	Information on more than 50 different data sources for hydrology
	UNOSAT Flood Dataset	Flood extent data, historical flood data, population and crop exposure statistics, etc., for selected flood events since 2007
	CEMS	Satellite images, meteorological data, etc. for near-real-time flood monitoring and forecasting
Text-Image	Global Flood Monitor	88 million tweets about more than 10,000 flood events across 176 countries in 11 languages
	CrisisMMD	Thousands of tweets and images collected during seven major natural disasters in 2017
	EM-DAT	Flood disaster records from 1900 to the present that meet the inclusion criteria

**Table 7 sensors-24-07090-t007:** Critical sensor types used in floods and their functions.

Floods	Sensors		Results
Pakistan Floods 2010	In Situ Sensors	River gauge stations	1. The scarcity of in situ sensors results in untimely and inaccurate flood warnings. 2. The limited grasp of climate-hydrology relationships and flood modeling techniques prevents effective data utilization. 3. The government’s inaccurate disaster assessment leads to untimely and inadequate relief efforts.
		Automatic weather stations	
	Weather Radar	Local weather radar stations	
	Satellite	Landsat-7, RADARSAT-2, ERS-2, Terra	
Thailand Floods 2011–2012	In Situ Sensors	Water level monitoring sensors, Weather sensors	1. Lack of historical sensor data for calibrating flood prediction outcomes. 2. Quarter of the country’s population has access to the Internet, but Internet usage dropped dramatically after the floods, leaving limited access to social media data. 3. Satellite imagery exhibits certain errors in urban area identification. 4. Damage to communication sensors impairs international rescue coordination.
	Weather Radar	Doppler weather radar stations	
	Social Media Sensors	Facebook, Twitter, YouTube, Flood reporting	
	Satellite	Landsat-7, RADARSAT-2, ALOS, ThaiChote, SPOT-5, COSMO-SkyMed, IRS	
Australia Floods 2011–2012	In Situ Sensors	Rainfall sensors, Water level sensors, Soil moisture sensors, Weather sensors, etc.	1. Sensors are effective, yet public disaster awareness is inadequate. 2. Facebook, the dominant social media data collection platform, played a key role in the early warning and rescue phase. 3. Extensive remote sensing data was obtained through international rescue efforts, leading to the development of localized flood management products.
	Weather Radar	Australian Weather Watch Radar Network	
	Social media Sensors	Facebook, Twitter, YouTube, Instagram, Blogs, Reddit	
	Satellite	Landsat-5, Landsat-7, RADARSAT-2, ALOS, Terra, Aqua, etc.	
Germany Floods 2013	In Situ Sensors	Water level monitoring sensors, Weather sensors	1. Multiple sensor data inputs to weather models for flood warning. 2. Utilizing UAVs in tasks such as dam breach monitoring reduces costs. 3. Government websites, social media platforms, and communication sensors effectively disseminate disaster information. 4. The timely activation of the International Charter facilitated the acquisition of remote sensing data from multiple countries.
	Weather Radar	DWD radar systems	
	Airborne Sensors	UAVs, Airborne radar	
	Social media Sensors	Twitter, Facebook, Google Maps, Länderübergreifendes Hochwasser Portal	
	Satellite	Landsat-8, TerraSAR-X, RADARSAT-2, ALOS Sentinel-2, Aqua, etc.	

## Data Availability

The collection of flood monitoring sensors data was uploaded for public download at github: https://github.com/bwtian/sensors-24-07090/blob/main/Flood%20monitoring%20sensors%20data%20collection.xlsx (accessed on 10 September 2024).

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
