# Peer review of "A Review of Cutting-Edge Sensor Technologies for Improved Flood Monitoring and Damage Assessment"

_sensors, 2024, doi:10.3390/s24217090_

Round 1
Reviewer 1 Report
Comments and Suggestions for Authors
Sensors-3222993
Title: A Review of Cutting-Edge Sensor Technologies for Improved Flood Monitoring and Damage Assessment
Main comments:
The review is to summaried the cutting-edge sensor technologies for improved flood monitoring and damage assessment. The paper is well writing and organizated. It conclused senors in three categories, but “AI and multi-source data integration” may need more details to deliver the conclusion.
I had difficult to read the table. It will be appreciated to add lines between different rows in the tables for better readability. The main findings are listed in tables, which need to be elaborated more in the writing to synziate the key information.
Minor comments:
Check spelling, e.g. P54 wanring; P158: during period of high, etc.
Line 115 ~119, “The floods can be xxxx management” can be revised to be more direct.
Figure 3 is confused and needs to be elaborated more to show why these keywords are important.
Figures 4 & 5 are difficult to read
Comments on the Quality of English LanguageOk, recommend with simple and direct writing style.
Reviewer 2 Report
Comments and Suggestions for Authors
Dear Authors,
The presented paper is highly valuable to the scientific community as it provides a comprehensive analysis of current solutions and limitations in flood assessment. The paper is well-written and meets high-quality standards. Here are my minor comments:
1. I suggest briefly discussing artificial intelligence algorithms for spatial data processing. The authors may consider referencing the papers on neural network applications with fused data from multispectral and radar sensing (https://doi.org/10.3390/rs15184463, https://doi.org/10.1016/j.isprsjprs.2021.08.016).
2. It would also be interesting to include a brief overview of available datasets for flood monitoring, such as Sen1Floods11 (https://openaccess.thecvf.com/content_CVPRW_2020/html/w11/Bonafilia_Sen1Floods11_A_Georeferenced_Dataset_to_Train_and_Test_Deep_Learning_CVPRW_2020_paper.html).
3. The authors may want to discuss why global solutions may not be effective in certain local cases and how such approaches should be customized.
4. Line 33: There is a typo: "developing counrris" should be corrected to "developing countries."
Line 173: The second bracket is missing: "(RFID."
Please also check for other typos throughout the manuscript.
Round 2
Reviewer 1 Report
Comments and Suggestions for Authors
improved significantly and addressed the comments very well.